# Insights into the Mechanisms of Heat Priming and Thermotolerance in Tobacco Pollen

**DOI:** 10.3390/ijms22168535

**Published:** 2021-08-08

**Authors:** Lavinia Mareri, Claudia Faleri, Iris Aloisi, Luigi Parrotta, Stefano Del Duca, Giampiero Cai

**Affiliations:** 1Department of Life Sciences, University of Siena, Via P.A. Mattioli 4, 53100 Siena, Italy; lavinia.mareri@unisi.it (L.M.); claudia.faleri2@unisi.it (C.F.); cai@unisi.it (G.C.); 2Department of Biological, Geological and Environmental Sciences, University of Bologna, Via Irnerio 42, 40126 Bologna, Italy; iris.aloisi2@unibo.it (I.A.); stefano.delduca@unibo.it (S.D.D.); 3Interdepartmental Centre for Agri-Food Industrial Research, University of Bologna, Via Quinto Bucci 336, 47521 Cesena, Italy

**Keywords:** pollen, heat priming, heat stress response, sugar metabolism, antioxidant response, calcium, cytoskeleton

## Abstract

Global warming leads to a progressive rise in environmental temperature. Plants, as sessile organisms, are threatened by these changes; the male gametophyte is extremely sensitive to high temperature and its ability to preserve its physiological status under heat stress is known as acquired thermotolerance. This latter can be achieved by exposing plant to a sub-lethal temperature (priming) or to a progressive increase in temperature. The present research aims to investigate the effects of heat priming on the functioning of tobacco pollen grains. In addition to evaluating basic physiological parameters (e.g., pollen viability, germination and pollen tube length), several aspects related to a correct pollen functioning were considered. Calcium (Ca^2+^) level, reactive oxygen species (ROS) and related antioxidant systems were investigated, also to the organization of actin filaments and cytoskeletal protein such as tubulin (including tyrosinated and acetylated isoforms) and actin. We also focused on sucrose synthase (Sus), a key metabolic enzyme and on the content of main soluble sugars, including UDP-glucose. Results here obtained showed that a pre-exposure to sub-lethal temperatures can positively enhance pollen performance by altering its metabolism. This can have a considerable impact, especially from the point of view of breeding strategies aimed at improving crop species.

## 1. Introduction

Pollen and pollen tubes are the male gametophyte of seed plants and are a key evolutionary step in the success of land plants [1]. Pollen activation and cytoplasmic polarization are fundamental for efficient pollen tube emergence and elongation, a prerequisite for fertilization [2]. Despite these processes involve countless proteins and ions, calcium is one the key players and its differential accumulation is fundamental to initiate cytoplasmic polarization and promote pollen tube growth [3]. The intracellular concentration of free Ca^2+^ is a balance between Ca^2+^ influx and Ca^2+^ sequestration, both extracellularly and inside organellar compartments. Ca^2+^, in combination with enhancers such as ROS, changes the dynamics of the cytoskeleton thus affecting intracellular transport [4]. Interactions between ion flux, such as Ca^2+^, ROS production, protein phosphorylation (and others not mentioned here) contribute to the polarization of pollen tubes and promote the secretion and deposition of new cell wall components in a finely- and timely-regulated manner [5,6]. Metabolism is also fundamental in cell wall deposition because it provides the activated monosaccharides for polysaccharide elongation (both pectin and cellulose) [7]. Among the many sugar-metabolizing enzymes, sucrose synthase (Sus) provides UDP-glucose, an energetic reservoir as well as the substrate for cellulose and callose synthesis [8].

Pollen tubes are very sensitive to high temperatures that can affect pollen viability, pollen germination as well as pollen tube growth. Initial stages of pollen development are more susceptible than later stages, which are relatively thermo-tolerant [9]. If pollen is subjected to excessive temperatures at the microspore stage, this leads to microspore abortion and to a drastic reduction in the number of pollen grains viable and capable of germinating [10]. Another key factor to consider in the heat stress response of pollen is the regimen of heat stress (HS) application. The ability of pollen to cope with a single episode of HS is known as basal thermotolerance (BTT), improvable if plant cells, prior to being exposed to acute HS, are subjected to a relatively high, non-lethal temperature. Alternatively, plant cells can experience a gradual temperature increase. In both cases, plant cells are subjected to the so-called “priming” or acquired thermotolerance (ATT) [10]. ATT has been applied as a pre-stress treatment only in tomato pollen and authors showed that heat pre-treatment improved the acquisition of pollen ATT compared to the direct stress, probably thanks to an ethylene-driven signaling [11]. HS response involves the activation of specific transcription factors named heat shock factors (HSFs), that in turn lead to the synthesis of heat-responsive genes [12,13]. Among the latter, well known is the expression of heat shock proteins (HSPs), molecular chaperones that accumulate in the cytoplasm and organelles and are involved in the stabilization, resolubilization, and refolding of proteins [14,15]. The expression and abundance of several HSFs and HSPs in developing anthers, microspores, and pollen subjected to HS has been widely characterized, e.g., HSP70, HSP90, and HSP100. Besides HSPs, also other proteins are potentially involved in the transduction of the HS response and cooperate to minimize cell damage. A large number of regulatory and functional stress-associated proteins have been recently reviewed and dehydrins and osmotins have been proposed to play a crucial role both in heat and drought stresses [16].

Another important alteration of pollen under HS concerns sugar metabolism. Under physiological conditions the vegetative cell is gradually filled with starch [17], that is converted into soluble sugars in maturing pollen. High temperatures alter starch accumulation and degradation in maturing pollen grains by changing either sugar transport or metabolism [17]. In addition to these modifications, mild HS also changes the allocation of soluble sugars to anther tissues [18]. Moreover, thermo-tolerant genotypes of sorghum, barley, soybean, and chickpea were shown to maintain high content of soluble sugars and showed higher pollen viability compared to sensitive genotypes [19,20,21]. It is also known that dehydrating pollen requires sucrose for the protection and preservation of native proteins and for the integrity of cellular membranes [11]. During heat stress, pollen and pollen tubes also cope with the unbalanced accumulation of reactive oxygen species (ROS). ROS are highly reactive metabolic by-products that act as signaling molecules, but they are toxic when exceeding their physiological concentrations [22]. Sophisticated ROS scavenging and detoxification mechanisms, consisting both in enzymatic (e.g., ascorbate peroxidase (APX), catalase (CAT), superoxide dismutase (SOD)) and non-enzymatic molecules (e.g., flavonoids) [23], are remarkably enhanced after heat treatment, both at transcriptional level and in activity. Although there are no direct evidences that ROS scavenging activity protects pollen development from ROS-induced damages, several mutant-based evidences highlight how the fine regulation of ROS content is essential for the production of viable pollen, indicating that increased scavenging activity might contribute to heat acclimation [24].

In the current work, we aimed to investigate the effects of ATT on pollen physiology and metabolism. In the first part of the work, we assessed the impact of priming on basic physiological parameters (e.g., pollen viability, pollen germination and pollen tube length) while in the second one we performed a multi-level analysis by studying Ca^2+^, ROS and related antioxidant systems, cytoskeletal organization, proteins, and sugar metabolism. Results herein obtained suggest that a pre-exposure to moderate stress (priming) can positively affect pollen overall performance by altering its metabolism. This can have a considerable impact, especially from the point of view of breeding strategies aimed at improving crop species.

## 2. Results

### 2.1. Pollen Viability, Germination, and Pollen Tube Elongation

The first data obtained concerned basic parameters of pollen and pollen tube. Analysis were conducted in 5 different groups: Control Group (C) maintained at room temperature (25 °C—RT) for the whole duration of the experiment; Primed Group (P) subjected to sub-lethal condition (1 h at 30 °C); Stressed Group (S) subjected to acute heat stress (2 h at 35 °C); Primed and Recovered Group (PR) subjected to priming and later to recovery (1 h at 30 °C and 3 h at RT) and finally the Primed, Recovered and Stresses Group (PRS) subjected to priming, recovery and stress (1 h at 30 °C, 3 h at RT and 2 h at 35 °C). The values of temperatures applied during the priming and stress phases were based on a previous work; please, see the Material and Methods section.

Figure 1A shows pollen viability no significant variations among treatments were detected with the only exception of S (acute heat stress) treatment where a significant decrease of pollen viability was observed. Interestingly, pollen subjected to priming (P) and priming plus recovery (PR) showed a pollen viability comparable to the control (C) indicating that the temperature used for priming was not able to affect this parameter. Notably, pollen subjected to both priming, recovery and finally stress (PRS) maintained unvaried viability.

Pollen germination (Figure 1B) and pollen tube length (Figure 1C) were also analyzed. After 1 h of germination, the pollen samples exhibited comparable values of germination, except for the PRS sample, which doubled the germination rate, and S sample that reduced this parameter. After 2 h of germination, the control sample (C) showed a germination percentage of over 40%, while the stressed sample (S) experienced a significant reduction in germination capacity (about 10%). While stress affected germination, priming (P) did not have the same effect since the germination rate was more than 30%; moreover, this value increased to 60% if priming was followed by recovery (PR). The primed and stressed samples (PRS) exhibited germination values just below the control but still significantly higher than the stressed sample (S). Prolonging germination to 3 h confirmed the results. Analysis of pollen tube elongation showed no significant differences among the investigated cases. Control pollen (C) showed a linearity of growth, with pollen tubes reaching about 250 μm in length after 3 h of germination. In all other cases, pollen tube length (either subjected to priming -P, or stress -S, or priming and recovery -PR, or priming, recovery and stress -PRS) was not significantly different from control. It can be deduced that pollen treated at moderate or elevated temperatures struggles to produce the pollen tube but the latter, once emitted, is still able to recover an adequate growth rate.

### 2.2. Kymograph Analysis of Pollen Tubes Revealed Differences among Experimental Groups

The kymograph analysis is an extremely convenient and fast way to analyze the growth profile of pollen tubes. Control samples (C) showed a linear growth profile, with average velocities about 1.4 μm/min. Velocities were very constant and the step between two fast growth peaks was in the order of 200–250 s (Figure 2A). In the case of pollen subjected to priming and germination, the speed was still comparable to the control (1.3 μm/min) but the step between two fast growth peaks was significantly longer, around 500–600 s (Figure 2B). In the case of stressed pollen (S), the growth velocity of pollen tubes was still relatively constant (around 1.5 μm/min). However, in this case the step between two fast growth peaks was higher than the control, about 300–400 s (Figure 2C). The primed and recovered sample showed a growth velocity comparable to the control and the step between two fast growth peaks was also similar to the control (Figure 2D). The sample subjected to priming, recovery, and stress (PRS) also behaved much the same as the control, both in terms of speed (1.4 μm/min) and in terms of the pitch between two fast growth peaks (Figure 2E).

### 2.3. Stress-Related Proteins Do Not Vary among Pollen Samples

Heat stress response is characterized by a deep rearrangement of protein synthesis. Molecular chaperones are synthesized to reestablish protein homeostasis, which is a prerequisite for cells to acquire ATT. In this study, we focused on osmotin and dehydrins that are proteins known to be involved in abiotic stress response, and on HSP70 that is a molecular chaperone (Figure 3). An anti-osmotin antibody shows that an 80-kDa cross-reacting protein was present at comparable levels in all samples with a slight increase in PR sample. In the case of dehydrin, immunoblotting analysis revealed two isoforms with different molecular weight: a high molecular weight protein of 65-kDa and a low molecular weight dehydrin of 20-kDa. Notably, their content was different, with the 65-kDa dehydrin less represented than the 20-kDa dehydrin. The latter showed a decline in P, S and PRS samples and an increase in PR samples. HSP70 did not appear to vary significantly between treatments, indicating that both priming and stress did not affect the expression of this protein. Figure 3B is a graph representing the blot intensities of osmotin and dehydrins compared to that of actin, while Figure 3C reports the ratio of signal intensity of HSP70 against actin.

### 2.4. The Content of Main Sugars Changes at Specific Treatments

To get a first assessment of the metabolic state of pollen tubes under the different heat treatments, the concentrations of the main sugars (sucrose, fructose, glucose, and UDP-glucose) were measured in pollen tubes. As can be seen in Figure 4A, the relative sucrose concentration did not change significantly after treatments. On the contrary, the content of both glucose and fructose only changed significantly after priming and heat stress (PRS). The data indicate that priming followed by recovery induces a significant increase in sucrose, which then presumably replenishes fructose and glucose levels under heat stress conditions. The two monosaccharides are actively used under stress conditions (PRS) to counteract the negative effects of heat stress, suggesting that one mechanism by which pollen becomes tolerant is the substantial accumulation of sucrose as induced by priming and recovery.

In addition to the three main sugars described above, we also analyzed the relative concentration of UDP-glucose (Figure 4B). The latter also derives from sucrose metabolism when sucrose is cleaved by the enzyme sucrose synthase. In addition, UDP-glucose is of relevance because it is the precursor of several cell wall polysaccharides. The highest value of UDP-glucose was observed in control (C). Both priming (P) and stress phase (S) induced a decrease in UDP-glucose levels. The addition, a recovery phase (PR) had no significant effect on UDP-glucose concentration. In contrast, priming, recovery, and stress treatment (PRS) induced a significant increase in UDP-glucose that regained the levels of primed samples.

Soluble pectins showed no significant differences among the various cases analyzed, indicating that heat treatments do not alter the release of pectins from the cell wall of pollen tubes (data not shown).

### 2.5. The Content of Sus Declines Following Priming and Stress while ATP Content Increases

Following UDP-glucose analysis, we considered a specific sugar-metabolizing enzyme, sucrose synthase (Sus; Figure 5), that provides UDP-glucose for cellulose and callose synthesis and that is an important hub in directing the metabolism of sucrose. Blotting analysis (Figure 5A) revealed that this enzyme is sensitive to heat treatment (both priming and stress) and that its content decreased when compared to control (C). To normalize the Sus content, the immunoblot intensity was correlated to the actin signal of same samples (Figure 5B). Sus content decreases already after priming but especially after priming and stress, indicating that raising temperatures (both 30 °C and 35 °C) can affect Sus levels.

Because sucrose can be cleaved and its products directed to glycolysis and respiration, we proceeded to determine the concentration of ATP. The latter is also an index of the energy requirements of cells. The concentration of ATP (Figure 6) is significantly increased in the primed samples (P) compared to the control (C). ATP content is barely detectable in samples subjected to stress (S), while it decreases significantly in samples subjected to priming and recovery (PR) as compared to the control. Of interest is that the ATP content of primed, recovered, and stressed samples (PRS) is comparable to that after priming and significantly higher than the control.

### 2.6. The Distribution of Actin Filaments Is Not Particularly Affected by Priming and Heat Stress

The organization of actin filaments was also checked (Figure 7). C, P, S, PR and PRS pollen was analyzed in ungerminated pollen (Stage a), when pollen tube is emitted (Stage b), when pollen tube length approximates the grain diameter (Stage c), and in longer pollen tubes (Stage d). The control pollen (C) sample exhibited a typical linear arrangement of actin filaments, uniformly distributed within the grain, not particularly focused towards specific areas. Primed pollen (P) exhibited no detectable alterations, with actin filaments arranged circularly within the pollen grain (Stage a). Stressed (S) pollen at Stage a often showed fluorescent punctuation, indicating potential damage or depolymerization of actin filaments (Figure 7 arrows). PR ungerminated pollen showed a homogeneous distribution of actin filaments, with no evidence of damage; actin filaments were apparently shorter and more fragmented. In the PRS ungerminated pollen, actin filaments were similar to C pollen, being sufficiently linear and widely distributed within the pollen grain, without damage. When pollen tube emerges (Stage b), actin filaments converged towards the aperture of the pollen grain concomitantly with the emission of the pollen tube (C). This organization was also clear in the primed sample (P) while it is less evident in the stressed and then germinated sample (S). In this case, actin filaments, although abundant, still appeared disorganized and not focused on the grain aperture. The focusing of actin filaments was more evident in samples subjected to priming followed by recovery, i.e., PR. In the latter, actin filaments were linearly organized from the grain to the pollen tube. During pollen tube elongation (Stage c), actin filaments were distinctly observable inside the pollen tube, predominantly arranged longitudinally with an evident disorganization at the apex (C). No visible alterations were highlighted in P, showing actin filaments comparably arranged with C. At this stage, actin filaments were comparable with the control sample also in S, PC and PRS. PR and PRS showed predominant abundance of actin filaments within the pollen grain, continuing within the pollen tube. No difference was shown in actin filament organization between samples when pollen tubes grew for longer times (Stage d). In fact, actin filaments were distinctly detectable in pollen tubes, arranged longitudinally with evidence of partial disorganization at the apex. Based on these results, we can affirm that although heat stress induced some damage to actin filaments in the pollen grain, actin filaments exhibited normal organization when pollen tubes grew. Pollen samples subjected to priming did not exhibit any damage.

In addition to considering actin distribution within pollen and pollen tube, we also analyzed the content of tubulin and of two specific post-translational modifications of tubulin (acetylation and tyrosination). Western blotting analysis (Figure 8A) did not show any changes in tubulin content for all samples. As for modified tubulin, levels of tyrosinated tubulin did not change between samples and acetylated tubulin was not detected. Detection of actin is also reported in the blot panel. The relative quantification of tubulins against actin is shown in Figure 8B, clearly showing that tubulin levels did not change after treatments. Thus, both tubulin content and any post-translational modifications (at least acetylation and tyrosination) do not correlate with tolerance or susceptibility to heat stress in pollen.

### 2.7. Proper Distribution of ROS in Pollen Tubes Is Restored Following Priming

Because of their importance in pollen tube growth and in response to stress or damage, ROS were monitored for both relative concentration and distribution. In control samples (Figure 9A), the typical distribution of ROS in pollen tubes was observed with a marked increase in the apical region, as shown by the red color. In primed pollen tubes (Figure 9B), ROS distribution was altered, although a faint accumulation could be seen at the apex. Stressed pollen tubes showed a very homogeneous ROS distribution, and the tube apex did not accumulate ROS (Figure 9C). The pattern and relative concentration of ROS in pollen tubes subjected to priming and recovery was similar to the priming case, albeit with a slight accumulation in the apical region (Figure 9D). A significant increase in ROS in the apical region was found in pollen tubes subjected to priming, recovery, and stress, albeit the apical accumulation was less focused than in control samples (Figure 9E). These observations suggest that single heat treatment (priming or stress) alters the profile of ROS in the pollen tube. In contrast, priming and recovery followed by stress restores the distribution of ROS.

### 2.8. Stress and Priming Affect Pollen Antioxidant Machinery

Changes in ROS distribution prompted us to the analysis of the pollen antioxidant machinery. Specific activities of the antioxidant enzymes superoxide dismutase (SOD) and catalase (CAT) were measured in crude extracts. SOD activity was pronounced in control pollen, with a value of 0.52 mU/mg of crude protein and stress significantly affected the enzyme activity, which dropped to 0.39 mU/mg of crude protein (S). SOD activity was not significantly altered in primed pollen (P). Conversely, the enzyme activity was less enhanced in pollen subjected to priming followed by recovery (PR) and in primed and recovered pollen subjected to stress (PRS), which showed SOD activities similar to pollen subjected to stress (S) (Figure 10A). CAT activity was also analyzed. Control pollen showed a mean CAT activity of 0.22 mM H_2_O_2_ min^−1^g^−1^, which decreased in stressed pollen (0.14 mM H_2_O_2_ min^−1^g^−1^). Mirroring SOD activity pattern, also for CAT activity, primed pollen (P) showed no significant differences with control pollen, while both pollen subjected to priming followed by recovery (PR) and primed and recovered pollen subjected to stress (PRS) showed a decreased CAT activity (Figure 10B).

### 2.9. Distribution of Cytosolic Ca^2+^ Changes upon Heat Treatment

As the tip-focused Ca^2+^ gradient is strongly associated with pollen tube emergence and growth, it was investigated by the Fluo-4/AM probe. In the control sample (C), the typical accumulation of cytosolic Ca^2+^ at the apex of pollen tubes was observable (Figure 11A). Priming induced alterations of the Ca^2+^ gradient, which impaired Ca^2+^ focus at the tube apex (Figure 11B,C). In the S sample, Ca^2+^ localized along the pollen tube but did not accumulate in the apical region. Two distinct experimental cases were observed. In one case, pollen tubes were characterized by high diffuse levels of cytosolic Ca^2+^, while in other cases a diffuse but lower Ca^2+^ content was observable (Figure 11D,E). Ca^2+^ distribution in the primed and recovered sample (PR) resembled the control condition, with a gradient along the pollen tube and a pool in the apex (Figure 11F,G). The Ca^2+^ gradient was visible also in the PRS sample, although fainter compared to control (Figure 11H,I).

## 3. Discussion

In this manuscript, we investigated how tobacco pollen can memorize a mild heat treatment (priming) and effectively respond to more drastic conditions. The data obtained demonstrate that various biochemical processes can be modulated by priming. These include Ca^2+^ concentration, proton gradient, and levels of reactive oxygen species. Soluble sugar (sucrose) content can also be modulated by heat pretreatment. All this results in a more effective pollen response (mainly proper tube growth) when subjected to heat stress. The choice of pollen as a model of study relies on the critical need for adaptation to HS during microsporogenesis and microgametogenesis. To date, thermotolerance is well studied at the plant level but to a lesser extent for pollen. Moreover, while several studies examined the response to heat stress in pollen, there is a lack of knowledge about pollen response to HS preceded by pre-adaptation [25]. The complexity of thermotolerance in pollen is emphasized by proteomics studies highlighting the importance of metabolic pathways [26], reinforcing the hypothesis that thermotolerance in a simple structure such as pollen grain or tube is multifaceted. Indeed, the implementation of various mechanisms, including hormones [11], as well as alternative splicing [27] or production of specific miRNAs [28], should be kept in mind if analyzing more complex systems.

At the cell level, one of the most perceptible effects of heat damage is the reduced germination of pollen grains as well as the reduced growth rate of pollen tubes, according to evidence since the early 1990s [29]. Due to the simplicity of measurements, the evaluation of germination and growth of pollen tubes is a reliable and useful approach for comparing distinct genotypes or different experimental conditions. While HS reduced germination rates also in the model system herein proposed, i.e., tobacco pollen, priming was shown to counteract the detrimental effect of HS. Interestingly, HS did not affect pollen tube growth, suggesting that the onset of germination is the most susceptible stage to HS. This evidence also hinted at the involvement of a series of buffering systems, the reason why we further investigated biochemical and metabolic processes allowing pollen adaptation.

First, evidence concerning the mitigation of stress by priming regarded the distribution of Ca^2+^ and ROS, both fundamental for proper growth of pollen tubes, and characterized by oscillatory concentrations related to growth rates [30,31,32,33,34]. Our results indicated that HS caused a sharp drop in the concentration of both Ca^2+^ and ROS, while priming had a positive effect on the redistribution of Ca^2+^ and ROS and contributed to their re-establishment at levels comparable to control. This tip-focused Ca^2+^ is well-known to crucial for pollen tube growth and orientation; and its influx likely derives from Ca^2+^ stores in the cell wall, via stretch- or voltage- or ROS-activated channels [34,35]. The maintenance of low Ca^2+^ concentrations just below the apex involves Ca^2+^-ATPase that actively either pumps Ca^2+^ externally or compartmentalizes it within cell organelles [36]. This dynamic Ca^2+^ balance requires a constant consumption of ATP, therefore investigated in our study. We found a decrease in ATP content, which may cause a redistribution of Ca^2+^ with a consequent effect on pollen tube growth. Concomitantly, ROS production involves Ca^2+^-dependent enzymes, underlying a deep interconnection among ATP, ROS and Ca^2+^ concentrations. Like Ca^2+^, also ROS accumulate in the apical region although their exact relationship with the growth process is not fully known [33]. Together with Ca^2+^ and pH gradient, ROS are likely part of the central oscillator that regulates the growth rate of pollen tubes [37]. In our analysis, we highlighted that HS-induced changes in Ca^2+^ and ROS content potentially resulting in an altered pattern of pollen tube growth. The typical growth oscillation, i.e., the switching between fast and slow growth, is lost after HS but is partially recovered if pollen tubes are previously exposed to priming. In addition, ROS-scavenging antioxidant enzymes, i.e., SOD and CAT, showed activities in primed samples comparable to control pollen, confirming that acclimation is mediated, at least in part, by the enhancement of cellular mechanisms preventing oxidative damage under stress [38,39]. Contrarily, after heat stress SOD and CAT activity decreased, partially confirming the data found in *Corylus avellana* pollen genotypes, in which HS reduced SOD and CAT activities [39].

Interestingly, also osmotin, a multifunctional protein that is overexpressed under stress conditions [40,41] and acts as an osmoprotectant, participates in the regulation of ROS level; the protein accumulates when pollen was stressed and continues to accumulate during recovery [42,43]. In our experimental setup, ROS content in S and PRS samples was lower than in P and PR samples, indicating that probably osmotin contributes to limit their concentration. However, the content of osmotin decreased under severe HS even if pollen previously received priming treatment, suggesting that the protein accumulated under sub-lethal temperatures might be used to cope with a more intense stress. The involvement of proteins of refolding systems besides osmotin, i.e., dehydrins and HSP70, in the acquisition of thermo-tolerance in tobacco pollen was also investigated because the so-called UPR (unfold protein response) is one of the main sensors of thermotolerance at the pollen level [44,45]. Like osmotin, dehydrins can regulate ROS content due to their high content in antioxidant amino acids, such as lysine, histidine, and glycine, and can scavenge ROS through oxidative modification. Two forms of dehydrins were detected in tobacco pollen, one remaining constant in all samples and the other one mirroring the osmotin profile progressively increasing in PR samples. HSP70 levels did not change between control, primed, and stressed samples, suggesting that the expression of HSP70 is not correlated to HS adaptation. This evidence might surprise; however, the role of HSPs in pollen response to HS is hypothesized although not definitively proven. In fact, several evidences show that HSPs (20, 22, 70, and 101) differentially accumulate in response to HS [46,47,48], but earlier studies by our group did not show a corresponding accumulation of HSP70 in tobacco pollen subjected to HS [49]. The molecular chaperone HSP 101 was proposed to co-operate with small HSP (sHSPs) and HSP70 chaperones to promote the removal of protein aggregates and it was shown to be essential to overcome HS [50]. Interestingly, phosphorylated sHSPs were proposed to interact directly or indirectly with F-actin, protecting actin filament against breakage caused by actin-severing proteins, and promoting its subsequent reorganization [51]. Since HSPs are conserved among living organisms, it would be interesting to deepen the roles of HSPs in the response to HS and stress adaptation.

Data reported in this work indicated that Ca^2+^ and ROS were altered in concentration and distribution after HS treatment and both may be involved in the organization of the cytoskeleton. To support this evidence, cytoskeleton organization, pollen tube growth, Ca^2+^ and ROS have recently been correlated in pear pollen [52]. The unaffected organization of the actin cytoskeleton after HS suggests that the flow of vesicles and organelles does not undergo profound alterations. Consequently, we can assume that the accumulation of secretory vesicles at the apex is undisturbed in heat-stressed tubes. Because we observed a major change in the growth pattern of pollen tubes, the secretory process may nevertheless be partially altered. Indeed, the growth at the pollen tube apex is the result of a balance between secretion of esterified pectins and subsequent conversion into acidic pectins [53]. The equilibrium between two different pectins is responsible for the oscillatory growth of pollen tubes, assuming that the turgor pressure is kept constant [54].

HS alters actin filaments and microtubules in terms of isoform accumulation and organization [49]. Therefore, the actin cytoskeleton was investigated, also in light of differences in ATP concentration, the latter involved in the assembly of cytoskeleton [55]. In tobacco pollen, under our experimental conditions, ATP levels decreased during HS but remained similar to control samples if pollen had been primed. These data indicate that pollen is able to produce ATP after priming and stress using a metabolic system that is not actuated by the stress condition only. As actin organization in both pollen and pollen tubes was regular and comparable to control pollen (apart from sporadic cases of depolymerized actin filaments), we might assume that both priming and HS do not have a prominent effect on actin filament organization. In parallel with the analysis of actin filaments, we also investigated the possible modification of and its modified isoforms, i.e., tyrosinated and acetylated tubulin, usually linked to distinct functions of microtubule subsets [56]. In tobacco pollen, no acetylated tubulin was detected, nor significant differences in tyrosinated tubulin among samples, suggesting that tolerance to HS does not involve post-translational modifications of tubulin. This is not surprising because the tubulin detyrosination/tyrosination cycle appears involved in regulating the transition of plant cells from elongation to division [57] and acetylated tubulin is sporadically present in pollen and anthers [58]. Unlike other plant cells and tissues, where acetylated microtubules are expectedly involved in kinesin-based regulation of motility [59], the presence and the role of acetylated microtubules in pollen is still an unanswered question.

Various sugars, in addition to being energy available for pollen tube growth, are also likely related to HS tolerance, and usually carbohydrate accumulation increases in response to HS conditions [60]. It must also be mentioned that carbohydrates are not the only metabolic mechanism of tolerance and lipids also play a role of some importance [61]. Understanding the role of carbohydrates requires knowledge of metabolic pathways. The levels of sugar did not show significantly differences in our analysis, although primed samples seemed to restore sugar concentrations comparable to the control sample. Therefore, we analyzed the levels of Sus, a key enzyme in sucrose-degrading metabolism [62]. Sus is active during pollen tube growth because it performs both energy-conserving action and produces activated substrates (UDP-glucose) for the synthesis of cellulose and callose [8,63]. In the case of heat-stressed tobacco pollen, Sus is already known to decreases significantly [49]. This manuscript confirmed that Sus content decreases in both primed and heat-stressed samples. However, the decrease in Sus levels is not necessarily a consistent trait of HS. For example, when HS occurs during pollen development, Sus transcripts increase in thermotolerant tomato genotypes [64], suggesting that Sus accumulation is an identifying trait of HS tolerance during microsporogenesis. Analyzing the levels of Sus and major related sugars, the lower accumulation of Sus during HS implies that the conservative energy pathway is affected, thus leading to lower levels of UDP-glucose. The rationale is that the breakdown of sucrose into UDP-glucose is of little use because pollen requires energy to withstand heat shock. It is, therefore, likely that sucrose is primarily degraded by invertases such that the products of sucrose degradation are redirected to the glycolytic and respiratory pathways for ATP synthesis. This hypothesis is confirmed by the fact that after priming and recovery, sucrose content increased significantly, suggesting an increased starch degradation activity or an increased sucrose uptake from the extracellular environment. At the same time, Sus content does not return to levels comparable to the control; therefore, the extra sucrose produced by priming and recovery will not be metabolized by Sus. Samples subjected to priming, recovery, and stress are characterized by a significant increase in UDP-glucose and simultaneously a significant decrease in glucose and fructose. We therefore assume that previously accumulated sucrose is cleaved into glucose and fructose, a fraction of which is used for the synthesis of new UDP-glucose. The rest of glucose and fructose is directed to ATP synthesis, whose content matches the control values.

In conclusion, our results show that HS significantly affects sugar levels, in particular sucrose, whose reduction presumably leads to a decrease in metabolic pathways. Among the various effects induced by the drop in available energy, we highlighted changes in Ca^2+^ content and distribution. These changes could explain the lower dynamics of actin filaments and thus the reduction in growth rate. Different accumulation of Ca^2+^ was interconnected with ROS levels, which was correlated to antioxidant enzymes levels. However, priming appears to be effective in rebalancing ROS and consequently Ca^2+^ concentration, which leads to restored growth conditions. Temperature is an essential parameter controlling crop yield and when HS occurs during the reproductive phase, plant fertility is affected, which reduces yield and affects the next generation of plants. Thus, so far, few molecular mechanisms responsible for low pollen germination frequency and distorted tube growth during stress conditions have been identified and, more importantly, mechanisms on how pollen responds to and circumvents stress limitation and how it adapts have been poorly studied. A major challenge for basic plant research will be to understand how plants adapt to heat stress to enable targeted plant improvements in the future. In particular, understanding the important molecular processes of thermoregulation in pollen may help identify “markers” for breeding new varieties with a less heat-sensitive fertilization process.

## 4. Materials and Methods

### 4.1. Pollen Growth and Stress Treatment

Pollen was harvested from plants grown in the greenhouse of the Botanical Garden (Department of Life Sciences, University of Siena) and subsequently dehydrated on silica gel, then stored at −20 °C. Before using, pollen was thawed and hydrated at room temperature (25 °C—RT) overnight in a moist chamber. Pollen was then harvested into Petri dishes and four experimental treatments were applied: Treatment 1 consisted in 1 h at 30 °C (Sample P); Treatment 2 consisted in 2 h at 35 °C (Sample S); Treatment 3 consisted in 1 h at 30 °C and 3 h at room temperature (Sample PR); Treatment 4 consisted in 1 h at 30 °C, 3 h at RT followed by 2 h at 35 °C (Sample PRS) (Figure 12). After treatments, pollen was germinated in BK medium supplemented with 12% sucrose [65]. All samples were compared to control pollen, namely pollen not subjected to heat treatment (Sample C). To determine pollen viability, the MTT (2,5-diphenyl tetrazolium bromide) test was used. MTT produces a yellowish solution that is converted to dark blue, water-insoluble MTT formazan by mitochondrial dehydrogenases of living cells. The test solution contained a 1% concentration of the MTT substrate in 5% sucrose. After 15 min incubation at 30 °C, the pollen samples were visualized under a light microscope. Pollen viability, germination rate and length of at least 100 pollen tubes were measured for all assays. The measurement of pollen tube lengths was carried out using the ImageJ software, after calibration. Following the required growth period, pollen was collected and used for additional analysis. The values of temperatures applied during the priming and stress phases were based on the work of Parrotta et al. [49].

### 4.2. Protein Extraction

After 1 h of germination, pollen tubes were collected by low-speed centrifugation and washed with HEM buffer (50 mM Hepes pH 7.5, 2 mM EGTA, 2 mM MgCl_2_) containing 12% sucrose. Pollen tubes were lysed in a cold room (4 °C) using a Potter-Elvehjem homogenizer (40 strokes); the lysis buffer was HEM supplemented with protease inhibitors and 1 mM DTT. Samples were centrifuged at 500× *g* for 10 min (4 °C). The supernatant was removed and centrifuged at high speed (100,000× *g* for 45 min at 4 °C). The resulting supernatant from the high-speed centrifugation was precipitated in 60% TCA in cold acetone for 1 h at −20 °C; precipitated proteins were washed in 100% cold acetone and then resuspended in 1-D electrophoresis buffer.

### 4.3. Determination of Protein Concentration

Protein concentration was determined using a commercial kit (2-D Quant Kit, GE HealthCare, Dornstadt, Germany), according to manufacturer’s instructions and using BSA as reference. Each sample was analyzed in three replicates.

### 4.4. 1-D Electrophoresis and Immunoblotting

Separation of proteins by 1-D electrophoresis was performed on precast 10% Criterion XT gels (Bio-Rad Laboratories, Segrate, Italy) using a Criterion cell (Bio-Rad Laboratories, Segrate, Italy) equipped with a Power Pac BioRad 300 at 200 V for approximately 45 min. Gels were stained with Bio-Safe Coomassie blue (Bio-Rad Laboratories, Segrate, Italy).

Transfer of proteins from gels to nitrocellulose or PVDF (for osmotin and dehydrins) membranes was performed using a Trans-Blot Turbo Transfer System (Bio-Rad Laboratories, Segrate, Italy) according to the manufacturer’s instructions. After blotting, membranes were blocked overnight at 4 °C in 5% ECL Blocking Agent (GE HealthCare Dornstadt, Germany) in 0.1% Tween-20 in TBS (20 mM Tris pH 7.5, 150 mM NaCl). After washing with TBS, membranes were incubated with the primary antibody for 1 h. The following primary antibodies were used: mouse monoclonal anti-tubulin B-5-1-2 (Sigma) diluted 1:5000, mouse monoclonal anti-actin 10B3 (Sigma) diluted 1:3000, rabbit polyclonal against maize sucrose synthase [66] diluted 1:10,000, mouse monoclonal against HSP70 (Euroclone Pero, Italy) diluted 1:5000, mouse monoclonal against the tubulin C-terminal tyrosine TUB-1A2 (Sigma-Aldrich St. Louis, MO, USA, diluted 1:1000), mouse monoclonal anti acetylated Lys 40 6-11B-1 (Sigma-Aldrich St. Louis, MO, USA) diluted 1:2000, rabbit monoclonal anti dehydrins 1:1000 and rabbit monoclonal anti osmotin 1:1000 (Agrisera Vännäs, Sweden). Subsequently, membranes were washed twice with TBS and then incubated for 1 h with peroxidase-conjugated secondary antibodies. Specifically, we used an anti-mouse IgG (Bio-Rad Laboratories, Segrate, Italy, diluted 1:3000) and a goat anti-rabbit IgG (Bio-Rad Laboratories, Segrate, Italy, diluted 1:3000). After rinsing the membranes with TBS, the immunological reactions were visualized with Immun-Star (Bio-Rad Laboratories, Segrate, Italy). Images of gels and blots were acquired using a Fluor-S apparatus (Bio-Rad Laboratories, Segrate, Italy) and analyzed with the Quantity One software (Bio-Rad Laboratories, Segrate, Italy). The quantification of the relative band intensity was normalized on actin and tubulin, used as housekeeping proteins as shown not to change in accumulation under our experimental conditions. Exposure times were 30–60 s for blots and 5–7 s for Coomassie-stained gels.

### 4.5. Fluorescence Imaging

For actin labeling, pollen tubes of tobacco were fixed and permeabilized in 100 mM Pipes pH 6.9, 5 mM MgSO_4_, 0.5 mM CaCl_2_, 0.05 % Triton X-100, 1.5 % formaldehyde, 0.05% glutaraldehyde for 30 min [67]. Samples were washed twice with the same buffer described above except that the pH was 7 and it contained 10 mM EGTA and 6.6 µM Alexa 543-phalloidin (Invitrogen Waltham, MA, USA). Samples were placed on slides and covered with a drop of Citifluor. At least 20 pollen tubes of comparable length were analyzed for each experimental condition.

For detection of reactive oxygen species (ROS), the fluorescent ROS indicator dye 2′,7′-dichlorodihydrofluorescein diacetate (DCFH2-DA; Molecular Probes Eugene, Eugene, OR, USA) was used. The detailed protocol is reported in [68]. Controls were carried out both without the ROS probe (in this case, we found no signal except for the autofluorescence of pollen grains) and by addition of fluorescein diacetate (in this case, we observed a signal uniformly diffused throughout the pollen grain and tube) (not shown). To highlight the differences in fluorescence intensity among different regions of pollen tubes, grayscale images were transformed into pseudocoloured images using the ImageJ software, specifically by the command Image > Lookup tables > 16 colors. At least 20 pollen tubes of comparable length were analyzed for each experimental condition.

Analysis of cytoplasmic Ca^2+^ was performed according to the protocol described in [69]. Briefly, pollen samples were incubated for 15 min at 4 °C with a solution of 0.5 μM of the Fluo-4/AM probe, 2% cetyl-trimethyl ammonium bromide (CTAB) in 100 mM Tris-HCl, 40 mM EDTA, pH 8.0. CTAB was used to increase the permeability of the cell wall to the probe. At least 20 pollen tubes of comparable length were analyzed for each experimental condition.

For all experimental assays, samples were observed with a Zeiss Axio Imager fluorescence microscope equipped with a 63× objective, an MRm AxioCam video camera and structured illumination. Images were captured and saved in zvi format.

To quantify the intracytoplasmic fluorescence signal of Ca^2+^ and ROS, individual images were imported into ImageJ and a segmented line was depicted to cover the ROI. The line thickness was such that it largely occupied the cell cytoplasm. The signal was measured with the Analyze > Plot profile command. Graphs showing Ca^2+^ and ROS profile are added in the Appendix A.

### 4.6. Kymograph Analysis

Video clips of pollen tubes under different experimental conditions were recorded to determine the growth profile for 1 h and 30 min after an initial 1-h germination in BK medium. Kymograph analysis reported a quantitative description of different measurements of pollen growth. Pollen was germinated in 24-well plates coated with polylysine to prevent pollen from moving during video recording. We used a Nikon phase-contrast inverted light microscope; video clips were captured with Pinnacle Media Center software (http://www.pctvsystems.com/, 11 May 2021) in MPEG-2 format. Next, videos were converted to AVI format using Virtual Dub software (http://virtualdub.org/, 11 May 2021), then opened in ImageJ and scanned with the kymograph plug-in (written by J. Rietdorf and A. Seitz; https://www.embl.de/eamnet/html/body_kymograph.html, 11 May 2021). For each frame, gray values were measured along a region of interest (ROI) manually specified by the operator. From the gray values, a new image (kymograph) was produced in which the X-axis is the time axis (unit is the interval between frames) and the Y-axis is the distance along the ROI (unit is the distance in pixels traveled by the apex of pollen tubes). The speed was measured by the same plug-in. Video clips were recorded for at least 10 pollen tubes in the various experimental cases.

### 4.7. Analysis of ATP and Sugars by High-Performance Liquid Chromatography (HPLC)

ATP was analyzed by HPLC (Perkin Elmer Milano, Italy, Series 200) following the method described by [70]. Fifty milligrams of pollen were collected by centrifugation at 135× *g* for 5 min from sucrose germination media and resuspended in 7% TCA (1 mL) as TCA solution stabilizes ATP half-life up to several hours (Sigma-Aldrich St. Louis, Missouri, USA bulletin). Complete disintegration and rupture of cells were performed with a Potter-Elvehjem homogenizer with 40 strokes per sample. The homogenate was centrifuged at 15,000× *g* for 15 min at room temperature. A total of 20 μL of each sample were injected into a solid stationary-phase C18 column (75 × 4.6 mm, particle size 5 μm). The mobile phase was a binary mobile phase gradient (Solvent A, 10 mm phosphate buffer pH 7; Solvent B, acetonitrile) with the following gradient: 0 min, 100% Solvent A, 0% Solvent B; 2 min, 95% A, 5% B; 4 min, 80% A, 20% B; 5.3 min, 75% A, 25% B; 6 min, 100% A, 0% B. The following parameters were used: flow rate 0.3 mL min^−1^; room temperature; approximate elution time 6 min for ATP. Identification of different components was obtained by programming a DAD 235C spectrophotometric detector with excitation wavelength 254 nm.

Sugar analysis was performed by lysing pollen with 1 mL of water; the final supernatants were examined by isocratic HPLC analysis with a Waters Sugar-Pak I ion-exchange column (6.5 × 300 mm) at a temperature of 90 °C and using a Waters 2410 refractive index detector. MilliQ grade water (pH 7) was used as a mobile phase with a flow rate of 0.5 mL min^−1^; an injection loop of 20 μL was used for all samples and standards (sucrose, glucose, fructose, and UDP-glucose).

### 4.8. Catalase and Superoxide Dismutase Spectrophotometric Assays

Germinated pollen was centrifuged for 5 min at 1000 rpm, room temperature, and pollen lysed as previously reported with minor modifications [68]. Briefly, pelleted pollen was suspended in PBS 150 mM, pH 7.5 with 0.1 mM EDTA (10 mg pollen/1 mL extraction buffer), sonicated twice on ice for 15 s and centrifuged for 25 min at 5000× *g*, 4 °C. Freshly prepared crude extracts were used to determine antioxidant enzyme activities according to plant-established protocols [71]. Total SOD activity was assayed in 48 well microplates (Corning, Corning, NY, USA) in 50 mM phosphate buffer (pH 7.8) containing 2 mM EDTA, 9.9 mM L-methionine, 55 μM NBT, and 0.025% Triton-X 100. Forty microliters of sample and 20 μL of 1 mM riboflavin were added and the reaction was started by light irradiation. The control plate was placed in the dark. Absorbance of the samples was measured by plate reader after 15 min at 560 nm (Infinite M Nano, Tecan, Switzerland). The enzyme activity (grams per fresh weight) of a sample was determined from a standard curve obtained by using pure SOD (Merk, Milano, Italy). For CAT, the decomposition of H_2_O_2_ was followed by a decrease in absorbance at 240 nm in a UV/Vis spectrophotometer (V530, Jasco Cremella (LC), Italy). The assay mixture contained crude extract diluted in 50 mM potassium phosphate buffer, pH 7.0 and 10 mM H_2_O_2_. The extinction coefficient of H_2_O_2_ (40 mM^−1^ cm^−1^ at 240 nm) was used to calculate the enzyme activity, expressed in terms of millimoles of H_2_O_2_ per minute per gram fresh weight.

### 4.9. Statistical Analysis

Pollen viability, germination rate and pollen tube length were analyzed using ImageJ software after calibration. For each treatment at least 100 pollen grains/tubes were considered, and experiment was conducted in triplicates. Differences between sample sets were determined by analysis of variance (one-way ANOVA, with a threshold *p*-value of 0.05).

## Figures and Tables

**Figure 1 ijms-22-08535-f001:**
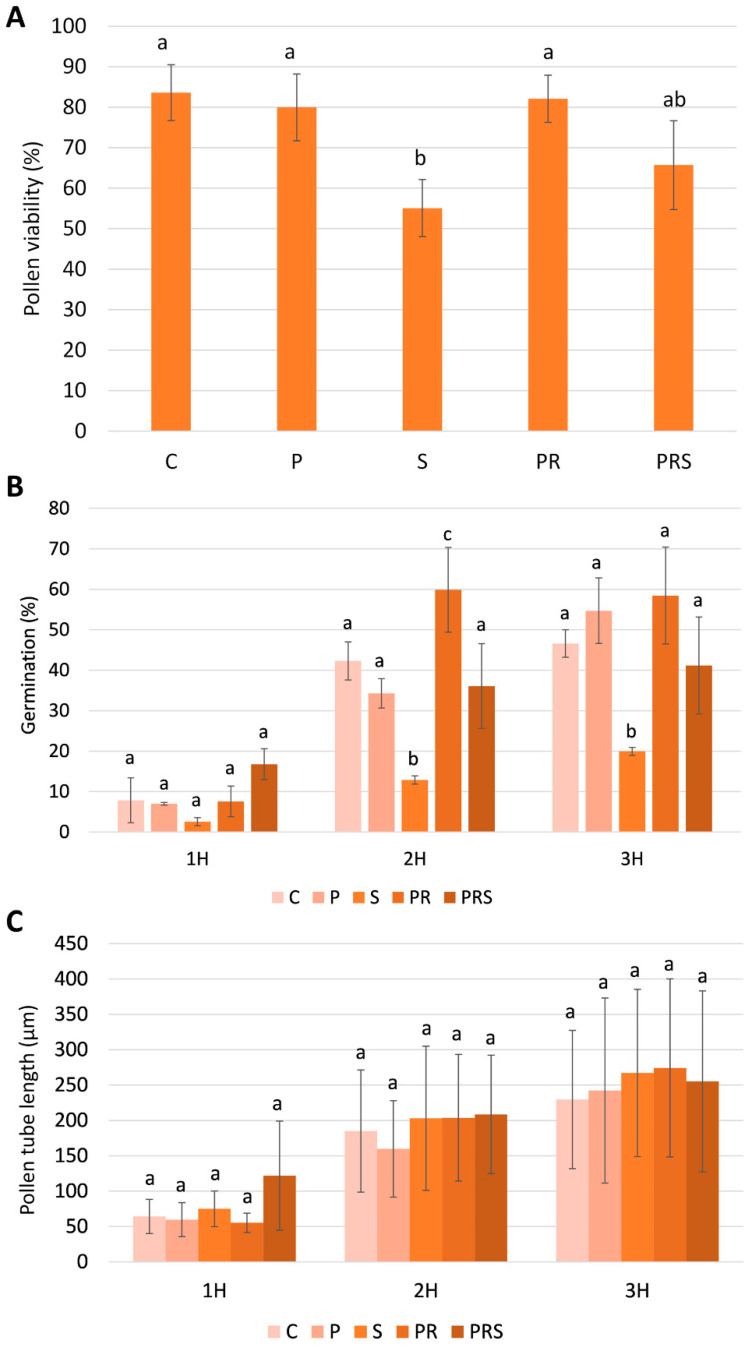
Analysis of pollen physiological parameters: (**A**) pollen viability; (**B**) pollen germination and (**C**) pollen tube length in control sample (C); primed sample (P); stressed sample (S); primed and recovered sample (PR); primed, recovered and stressed sample (PRS). For each treatment at least 100 pollen grains were considered, and results are expressed as averages ± standard deviation of three experiments analyzed in triplicate. Averages were compared by one-way ANOVA (for pollen germination and pollen tube growth means were compared within 1 h, 2 h and 3 h). Bars marked with the same letter are not significantly different (*p* ≥ 0.05).

**Figure 2 ijms-22-08535-f002:**
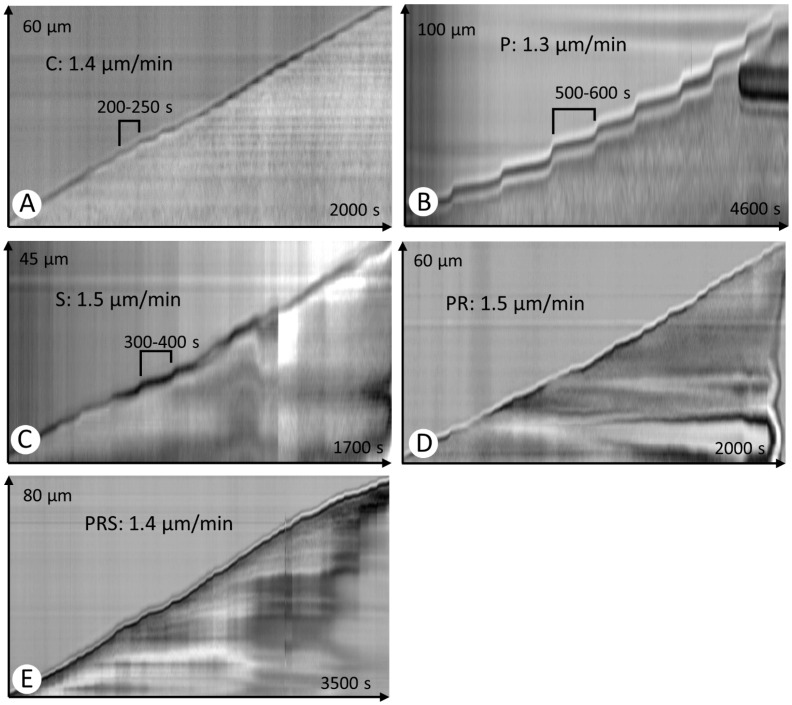
Kymograph analysis of pollen tubes under different experimental conditions. For each test condition, the *y*-axis shows the distance in µm, while the x-axis shows the analysis time in sec. Each image also shows the average value of growth velocity referred to the analysis time: (**A**) control sample (C). The step between two successive velocity peaks is indicated; (**B**) primed sample (P); (**C**) stressed sample (S); (**D**) primed and recovered sample (PR); (**E**) primed, recovered and stressed sample (PRS).

**Figure 3 ijms-22-08535-f003:**
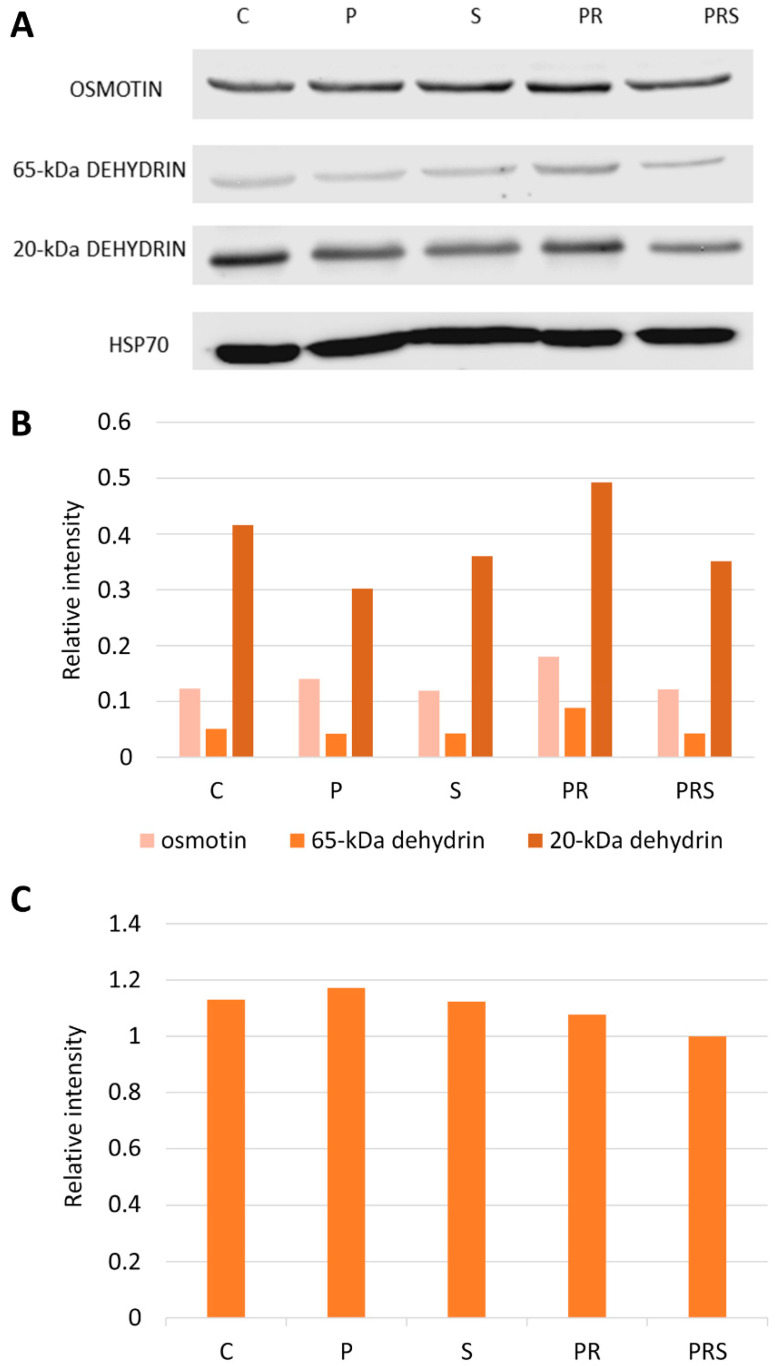
Immunoblotting with anti-osmotin, anti-dehydrin and anti-HSP70 antibody: (**A**) Detection of osmotin, dehydrin and HSP70 in the different pollen samples. Lane 1, control sample (C); Lane 2, primed sample (P); Lane 3, stressed sample (S); Lane 4, primed and recovered sample (PR); Lane 5, primed, recovered and stressed sample (PRS). A total of 30 µg of proteins was loaded in each lane; (**B**) measurement of osmotin, 65-kDa dehydrin and 20-kDa dehydrin immunoblot intensity; (**C**) Measurement of HSP70 blot intensity. In all cases, data were normalized against actin, chosen as the reference protein.

**Figure 4 ijms-22-08535-f004:**
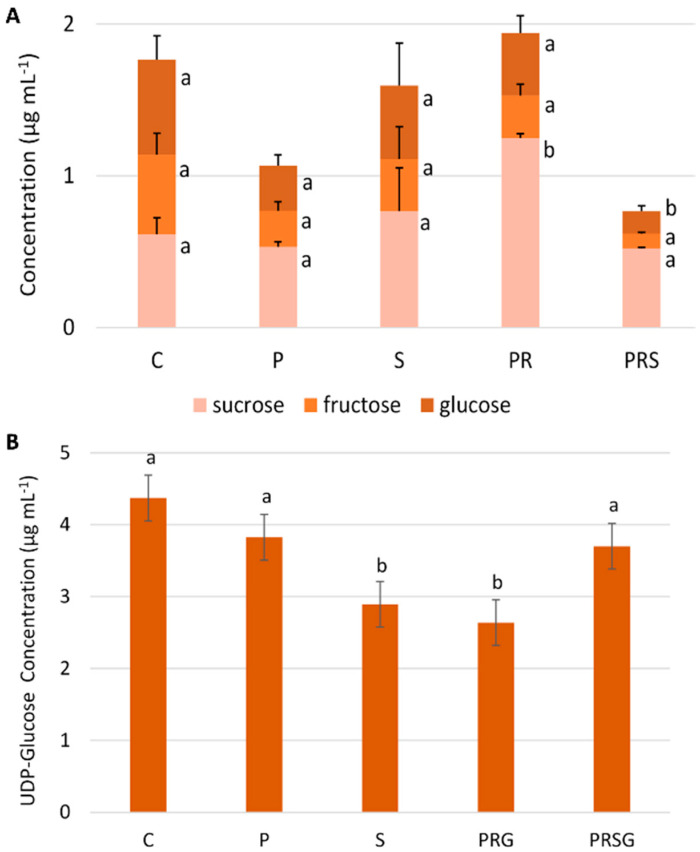
Content of main sugars in pollen samples: (**A**) concentration of sucrose, glucose, and fructose in control sample (C), primed sample (P), stressed sample (S), primed + recovery sample (PR) and primed + recovery + stress sample (PRS). Points with the same lower-case letters do not differ significantly (*p* > 0.05); (**B**) concentration of UDP-glucose. Concentration is expressed as µg/mL.

**Figure 5 ijms-22-08535-f005:**
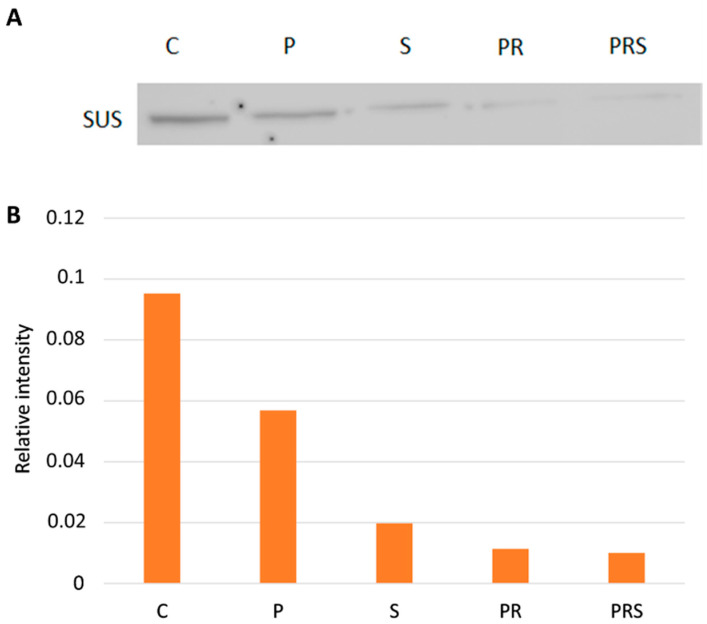
Immunoblotting with anti-Sus antibody: (**A**) Detection of Sus in different pollen samples. Lane 1, control sample (C); Lane 2, primed sample (P); Lane 3, stressed sample (S); Lane 4, primed and recovered sample (PR); Lane 5, primed, recovered and stressed sample (PRS). A total of 30 µg of proteins was loaded in each lane; (**B**) measurement of Sus immunoblot intensity. The relative content of Sus was normalized against the content of actin, considered as a reference protein.

**Figure 6 ijms-22-08535-f006:**
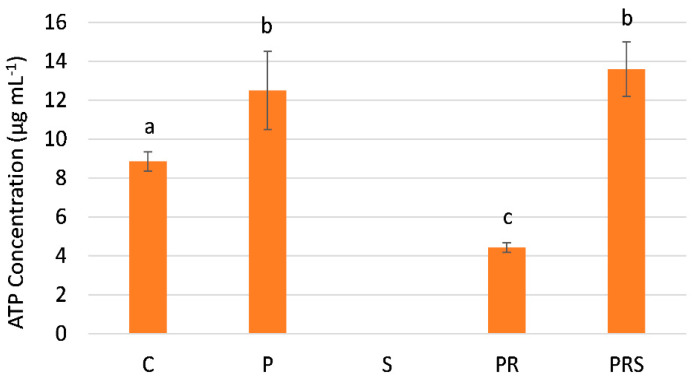
ATP concentration in control sample (C); primed sample (P); stressed sample (S); primed and recovered sample (PR); primed, recovered and stressed sample (PRS). Averages ± standard deviation of three experiments analyzed in triplicate are reported. Averages were compared by one-way ANOVA. Bars marked with the same letter are not significantly different (*p* ≥ 0.05).

**Figure 7 ijms-22-08535-f007:**
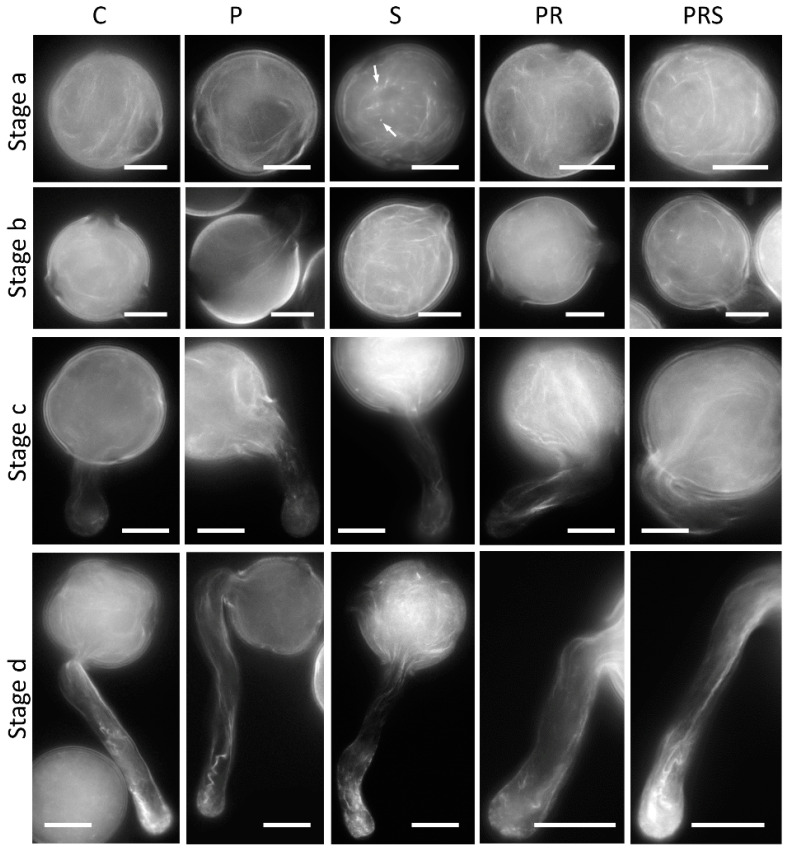
Distribution of actin filaments in control sample (C); primed sample (P); stressed sample (S); primed and recovered sample (PR); primed, recovered and stressed sample (PRS). Stage a: actin filaments in the ungerminated grain; Arrows indicate fragmented actin filaments. Stage b: emerging pollen tube stage; Stage c: pollen tubes with length comparable to the pollen grain diameter; Stage d: pollen tubes with increased length. Figure reports the most representative images. At least 30 pollen grains and tubes were analyzed. Refer to the main text for a more detailed description. Bars 20 µm.

**Figure 8 ijms-22-08535-f008:**
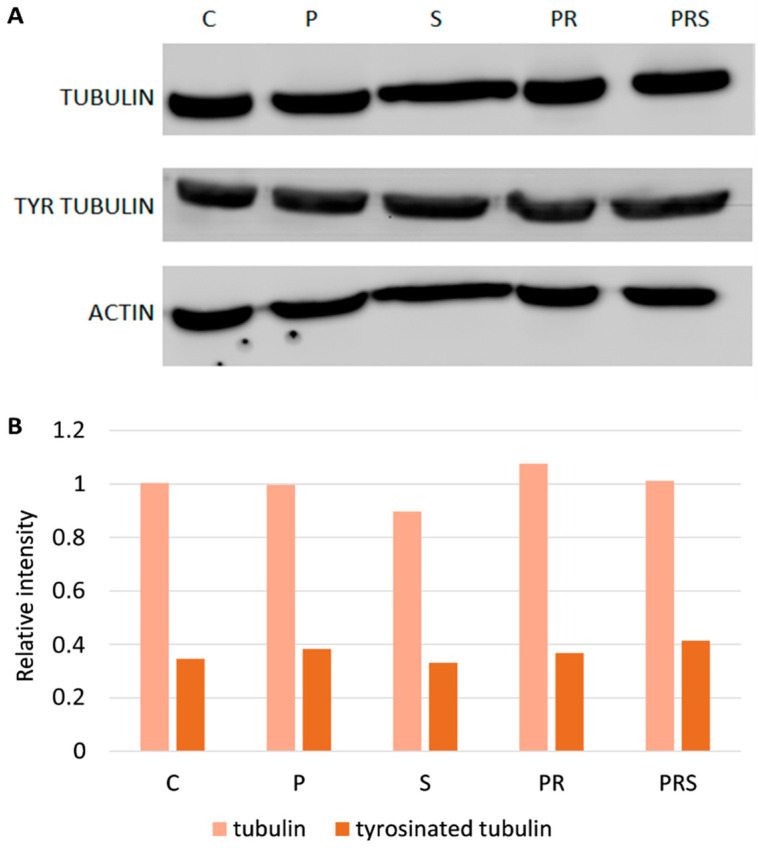
Detection of tubulins in pollen samples: (**A**) immunoblots with antibodies to tubulin, tyrosinated and acetylated tubulin as well to actin. Lane 1, control sample (C); Lane 2, primed sample (P); Lane 3, stressed sample (S); Lane 4, primed and recovered sample (PR); Lane 5, primed, recovered and stressed sample (PRS). A total of 30 µg of proteins was loaded in each lane; (**B**) relative quantitation of tubulin (TUB) and tyrosinated tubulin (TYR TUB). All samples were normalized using the actin signal.

**Figure 9 ijms-22-08535-f009:**
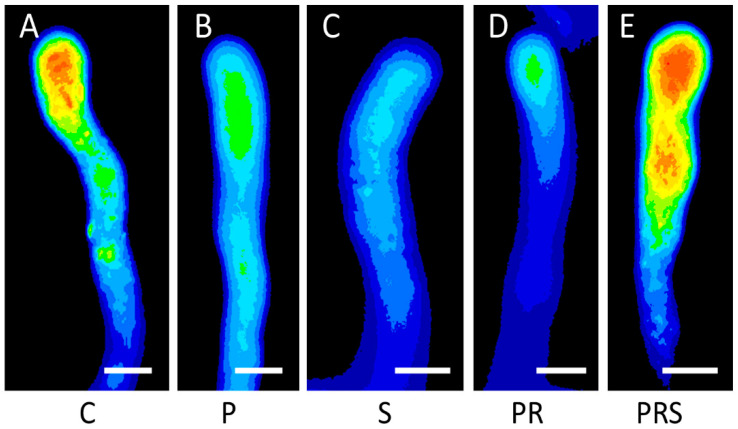
Analysis of reactive oxygen species (ROS) distribution in pollen tubes: (**A**) control sample (C); (**B**) primed sample (P); (**C**) stressed sample (S); (**D**) primed and recovered sample (PR); (**E**) primed, recovered and stressed sample (PRS). Figure reports the most representative images. At least 30 pollen grains and tubes were analyzed. Bars 10 µm.

**Figure 10 ijms-22-08535-f010:**
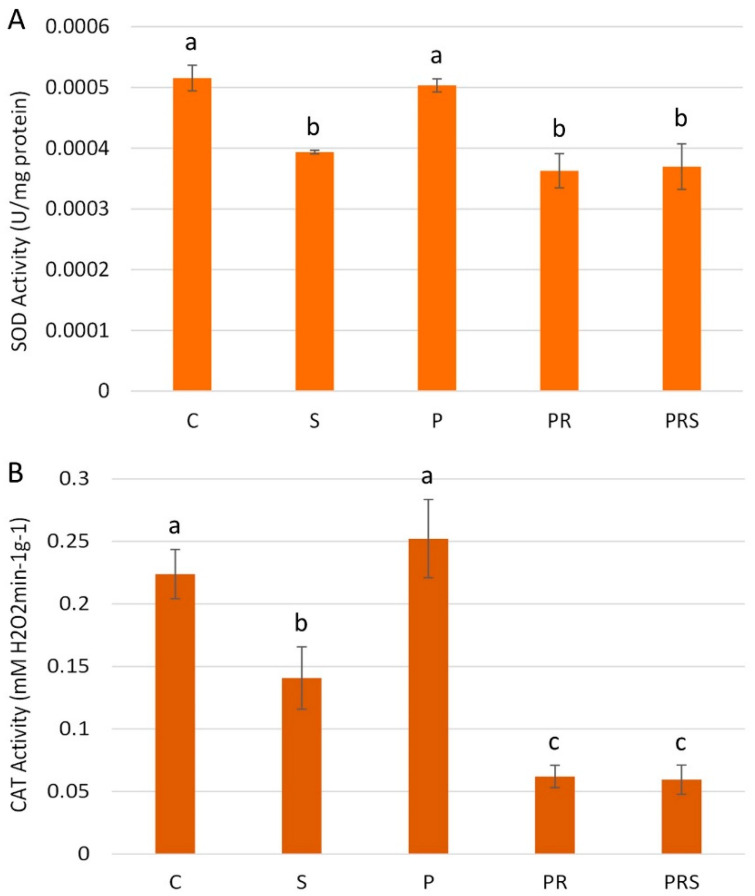
Analysis of pollen enzymatic machinery: (**A**) SOD activity and (**B**) CAT activity in control sample (C); primed sample (P); stressed sample (S); primed and recovered sample (PR); primed, recovered and stressed sample (PRS). Averages ± standard deviation of three experiments analyzed in triplicate are reported. Means were compared by one-way ANOVA. Bars marked with the same letter are not significantly different (*p* ≥ 0.05).

**Figure 11 ijms-22-08535-f011:**
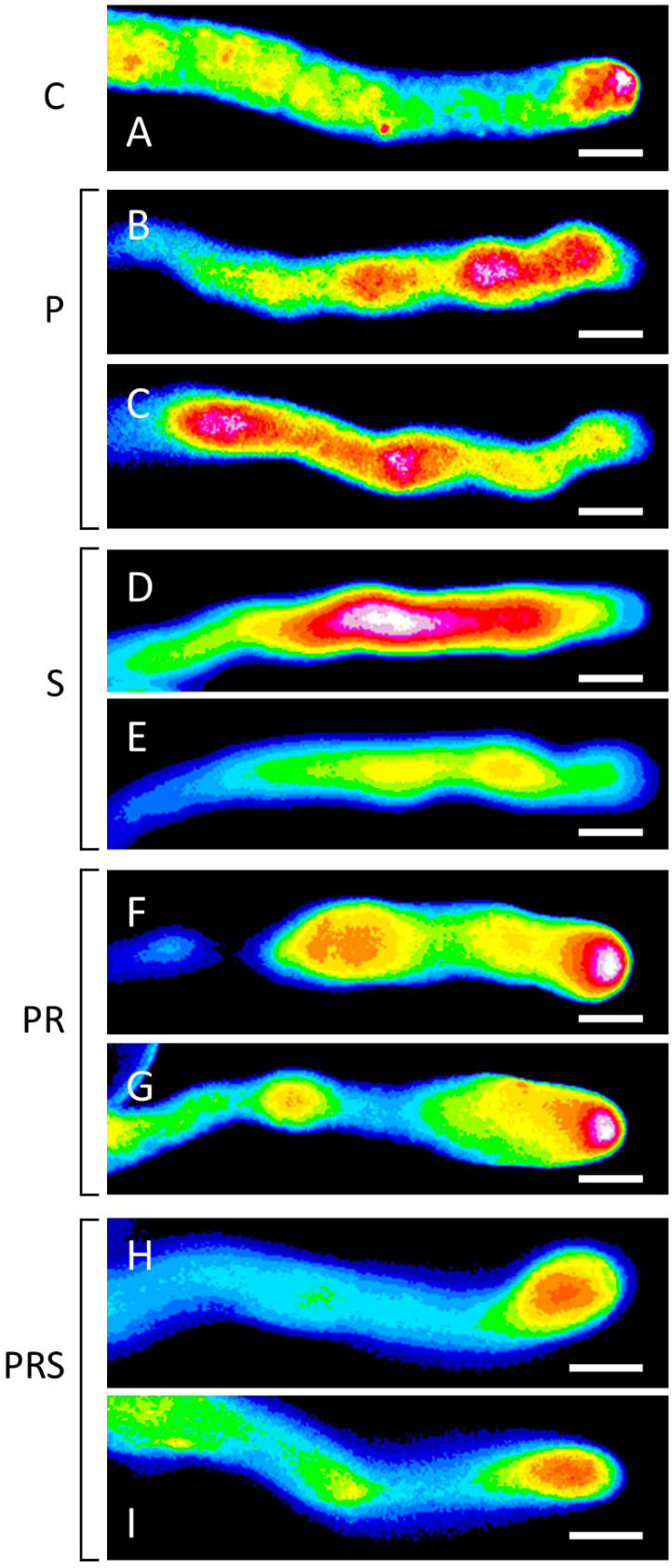
Distribution of cytosolic Ca^2+^ in pollen tubes: (**A**) control sample (C), with the typical accumulation of cytosolic Ca^2+^ at the pollen tube apex; (**B**,**C**) primed sample (P), with altered Ca^2+^ gradient not always focused at the tube apex; (**D**,**E**) stressed sample (S), with Ca^2+^ localized along the pollen tube but not in the apical region; (**F**,**G**) primed and recovered sample (PR), with diffuse signal along the pollen tube but more evident in the apical region; (**H**,**I**) primed, recovered and stressed sample (PRS), with Ca^2+^ signal again localized to the tube apex although with less intensity. Figure reports the most representative images. At least 30 pollen grains and tubes were analyzed. Bars 10 µm.

**Figure 12 ijms-22-08535-f012:**
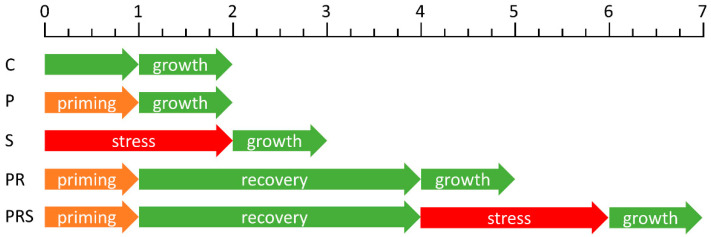
Diagram of heat treatments to which pollen was subjected (green: room temperature; orange: priming at 30 °C; red: stress at 35 °C). The scale at the top is the time in hours. The term “growth” indicates pollen tube germination.

## Data Availability

Data available on request due to restriction.

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
