# Peer review of "Insights into the Mechanisms of Heat Priming and Thermotolerance in Tobacco Pollen"

_ijms, 2021, doi:10.3390/ijms22168535_

Round 1

Reviewer 1 Report

Review of Manuscript ID: ijms-1318644 Type of manuscript: Article Title:
INSIGHTS INTO THE MECHANISMS OF HEAT PRIMING AND THERMOTOLERANCE IN TOBACCO POLLEN

Authors present a broad range of research how heat priming affects physiological parameters, biochemical stress markers such as Ca2+ level and ROS, HSPs levels, organization of cytoskeleton, and sugar metabolism in tobacco pollen and pollen tubes. I agree with Authors that thermotolerance or priming for male gametophyte is less studied. Therefore, global warming context, developmental priming and assays performed on model plant can provide the impact in the field.

The article is well written, objective and detailed. It is easy to follow every part of the article. The figures are high quality and informative. All assays are repeated and significance is well shown. The discussion is data-based and consistent with the findings.

I do not have major concerns about presented study.

Minor issues:

I could not find the information how viability test on pollen was performed?

I found double dots in line 187 and 245.

Comments:

As Authors show in manuscript surprisingly HSP 70 levels did not change during all experimental assays – in control, stressed, primed, and primed/stressed pollen. This was also mentioned in former work of Authors Parrotta et al (2016) concerning heat stress and pollen tubes.

I think it is interesting for future investigation to check other HSPs as markers for pollen and pollen tubes than HSP 70.

McLoughlin et al (2019) points:

https://academic.oup.com/plphys/article/180/4/1829/6117551

‘HSP101 belongs to the Hsp100/casein lytic proteinase (Clp) B family of AAA+ chaperones (ATPase associated with diverse cellular activities). The molecular chaperone heat shock protein (HSP) 101 is a protein disaggregase that co-operates with the small HSP (sHSP) and HSP70 chaperones to facilitate removal of such aggregates and is essential for surviving severe heat stress. ‘

Also Burke et al (2015):

https://journals.plos.org/plosone/article?id=10.1371/journal.pone.0122933

‘The AtHSP101 construct under the control of the constitutive ocs/mas ‘superpromoter’ was transformed into cotton Coker 312 and tobacco SRI lines via Agrobacterium mediated transformation. Thermotolerance of pollen was evaluated by in vitro pollen germination studies. Comparing with those of wild type and transgenic null lines, pollen from AtHSP101 transgenic tobacco and cotton lines exhibited significantly higher germination rate and much greater pollen tube elongation under elevated temperatures or after a heat exposure’

On the other hand, Chaturvedi et al (2016) show protein candidates for defence during developmental processes such as well-known HSP 70, but also HSP 20 and HSP 22.

https://pubmed.ncbi.nlm.nih.gov/27271282/

‘Several heat-shock proteins (HSP 20, HSP 22 and HSP 70) were detected with high levels which might not only protect pollen mother cells under heat stress condition but also prepare the cells to undergo meiotic and mitotic divisions during the next step in the developmental process.’

Mounier and Ariigo (2002) propose a model of the interaction actin-sHSPs:

https://www.ncbi.nlm.nih.gov/pmc/articles/PMC514814/

‘Model of the protection of actin microfilaments by sHsps: In unstressed cells sHsps form large aggregates of nonphosphorylated monomers. The earliest responses to heat shock or other stresses are phosphorylation of sHsps, disruption of sHsp large aggregates after, in some cases, a transient hyperoligomerization, and disorganization of the actin cytoskeleton. Phosphorylated sHsps organized in small oligomers would interact directly or indirectly with F-actin, protect the actin filament against breakage by actin-severing proteins, and promote its subsequent reorganization. Nonphosphorylated monomers may cap the plus end of the actin filament and participate in the regulation of the microfilament assembly’

Since HSPs are conserve among living organisms, it is very interesting whether unaffected organization of the actin cytoskeleton which Authors observed (only slightly fluorescent punctuation, indicating potential damage or depolymerization of actin filaments, mainly in stressed pollen) is correlated with protection of HSP 20.

I hope Authors find this information helpful maybe for the discussion part or further experiments.

Author Response

Review of Manuscript ID: ijms-1318644 Type of manuscript: Article Title:
INSIGHTS INTO THE MECHANISMS OF HEAT PRIMING AND THERMOTOLERANCE IN TOBACCO POLLEN

Authors present a broad range of research how heat priming affects physiological parameters, biochemical stress markers such as Ca2+ level and ROS, HSPs levels, organization of cytoskeleton, and sugar metabolism in tobacco pollen and pollen tubes. I agree with Authors that thermotolerance or priming for male gametophyte is less studied. Therefore, global warming context, developmental priming and assays performed on model plant can provide the impact in the field.

The article is well written, objective and detailed. It is easy to follow every part of the article. The figures are high quality and informative. All assays are repeated and significance is well shown. The discussion is data-based and consistent with the findings.

I do not have major concerns about presented study.

We thank the Reviewer for the accurate correction of our paper and for the useful suggestion reported below.

Minor issues:

I could not find the information how viability test on pollen was performed?

I found double dots in line 187 and 245.

We thank the Reviewer for remarking the absence of details about the viability assay. In the revision of the manuscript, we added some words about the basis of MTT viability test in M&M. Moreover, we added some details on the parameters chosen for calculating the viability after HS treatment. In particular, we specify the minimal number of counted pollens per treatment and the number of experimental replicates. We would like to underline that we used high-performing pollen (>45% germination, very homogeneous) for all the experiments reported in the paper.

Comments:

As Authors show in manuscript surprisingly HSP 70 levels did not change during all experimental assays – in control, stressed, primed, and primed/stressed pollen. This was also mentioned in former work of Authors Parrotta et al (2016) concerning heat stress and pollen tubes.

I think it is interesting for future investigation to check other HSPs as markers for pollen and pollen tubes than HSP 70.

McLoughlin et al (2019) points:

https://academic.oup.com/plphys/article/180/4/1829/6117551

‘HSP101 belongs to the Hsp100/casein lytic proteinase (Clp) B family of AAA+ chaperones (ATPase associated with diverse cellular activities). The molecular chaperone heat shock protein (HSP) 101 is a protein disaggregase that co-operates with the small HSP (sHSP) and HSP70 chaperones to facilitate removal of such aggregates and is essential for surviving severe heat stress. ‘

Also Burke et al (2015):

https://journals.plos.org/plosone/article?id=10.1371/journal.pone.0122933

‘The AtHSP101 construct under the control of the constitutive ocs/mas ‘superpromoter’ was transformed into cotton Coker 312 and tobacco SRI lines via Agrobacterium mediated transformation. Thermotolerance of pollen was evaluated by in vitro pollen germination studies. Comparing with those of wild type and transgenic null lines, pollen from AtHSP101 transgenic tobacco and cotton lines exhibited significantly higher germination rate and much greater pollen tube elongation under elevated temperatures or after a heat exposure’

On the other hand, Chaturvedi et al (2016) show protein candidates for defence during developmental processes such as well-known HSP 70, but also HSP 20 and HSP 22.

https://pubmed.ncbi.nlm.nih.gov/27271282/

‘Several heat-shock proteins (HSP 20, HSP 22 and HSP 70) were detected with high levels which might not only protect pollen mother cells under heat stress condition but also prepare the cells to undergo meiotic and mitotic divisions during the next step in the developmental process.’

Mounier and Ariigo (2002) propose a model of the interaction actin-sHSPs:

https://www.ncbi.nlm.nih.gov/pmc/articles/PMC514814/

‘Model of the protection of actin microfilaments by sHsps: In unstressed cells sHsps form large aggregates of nonphosphorylated monomers. The earliest responses to heat shock or other stresses are phosphorylation of sHsps, disruption of sHsp large aggregates after, in some cases, a transient hyperoligomerization, and disorganization of the actin cytoskeleton. Phosphorylated sHsps organized in small oligomers would interact directly or indirectly with F-actin, protect the actin filament against breakage by actin-severing proteins, and promote its subsequent reorganization. Nonphosphorylated monomers may cap the plus end of the actin filament and participate in the regulation of the microfilament assembly’

Since HSPs are conserve among living organisms, it is very interesting whether unaffected organization of the actin cytoskeleton which Authors observed (only slightly fluorescent punctuation, indicating potential damage or depolymerization of actin filaments, mainly in stressed pollen) is correlated with protection of HSP 20.

I hope Authors find this information helpful maybe for the discussion part or further experiments.

We are really grateful for the suggestion of implementing bibliography and we acknowledge the usefulness of the suggested references not only for improving the manuscript, but also for planning future research. For this reason, we opted for not extending excessively the discussion; however, some recent references have been added in the text in order to make it more comprehensive as suggested by referee.

Reviewer 2 Report

In this manuscript entitled “Insights Into the Mechanisms of Heat Priming and Thermotolerance in Tobacco Pollen”, Mareri et al reported that a pre-exposure to sub-lethal temperature can positively affect pollen performance. It was a crucial questions and worthy to be investigated, due to global warming and the thermal sensitivity of male gametophyte. This research aimed to evaluate basic physiological parameters and the related aspects on pollen functions, such as, calcium level, Reactive Oxygen Species (ROS), related antioxidant systems, the organization of actin filaments and cytoskeletal protein, as well as sucrose synthase concentrations. This manuscript was organized well and easy to follow. I have some concerns to be addressed. 

  1. Figure 3A, Does it seem that HSP70 is immunoblotted as control? If so, the blots showed overexposure. It would be better to replace a less exposed blot for qualification. The same revisions are needed on Figure 8A, tubulin, TYR tubulin, and actin blots. 
  2. Figure 5A, is it possible to add loading control for qualification? For Figure 3, 5, and 8, it would be better to indicate which blots are loaded as control in figure legends.

Author Response

In this manuscript entitled “Insights Into the Mechanisms of Heat Priming and Thermotolerance in Tobacco Pollen”, Mareri et al reported that a pre-exposure to sub-lethal temperature can positively affect pollen performance. It was a crucial questions and worthy to be investigated, due to global warming and the thermal sensitivity of male gametophyte. This research aimed to evaluate basic physiological parameters and the related aspects on pollen functions, such as, calcium level, Reactive Oxygen Species (ROS), related antioxidant systems, the organization of actin filaments and cytoskeletal protein, as well as sucrose synthase concentrations. This manuscript was organized well and easy to follow. I have some concerns to be addressed. 

We are grateful to the reviewer for the particular attention to our paper. We checked the whole manuscript and added details whenever necessary; we paid specific attention to the addressed concerns.

Figure 3A, Does it seem that HSP70 is immunoblotted as control? If so, the blots showed overexposure. It would be better to replace a less exposed blot for qualification.

The same revisions are needed on Figure 8A, tubulin, TYR tubulin, and actin blots. 

Figure 5A, is it possible to add loading control for qualification? For Figure 3, 5, and 8, it would be better to indicate which blots are loaded as control in figure legends.

We acknowledge that the membrane of HSP70 is overexposed and we recognize this might lead to a misquantification. The problem is that the accumulation of HSP70 is huge; given the high specificity of the antibody, it is technically difficult to reduce overexposition (even at very short acquisition times), as the quantity of loaded proteins should be kept high in order to detect the signal of other proteins (such as sucrose synthase). We had to choose between performing the analysis under the same conditions or making a separate blot only for HSP70, but loading a lower quantity of protein extract. We have chosen to perform all immunoblottings and analysis on the same membrane, thereby assuring highest confidence and reproducibility of the experiments.

About the quantification/normalization of band intensities, actin was always chosen as housekeeping protein, as its accumulation did not change in our experimental conditions. For this reasons, band intensity was normalized to actin.

Reviewer 3 Report

Mareri et al., Insights Into the Mechanisms of Heat Priming and Thermotolerance in Tobacco Pollen

In this paper, the authors used tobacco pollens to study the effects of heat priming and heat stress on pollen viability, germination, pollen tube length, pollen viability, Calcium level, ROS, organization of actin filaments,  cytoskeletal proteins, sugars, and sucrose synthase.  This is a very detailed study with different aspects related to pollen thermotolerance. As the authors mentioned, the effects of priming on pollen grains have been rarely studied. Thus, the data presented here are an important contribution to the thermal tolerance of pollen and have important implications for improving plant thermal tolerance. The manuscript is also generally well-written. In some places, the clarity of the writing could be improved. For example, it is unclear in the discussion that which are the results/findings from the present study, and which are from the literature.  So I would suggest clarifying those. The language could also be carefully checked by a native English speaker. The following are some detailed comments:

Line 49: Cite references to support the statement that pollen tubes are very sensitive to high temperatures.

Line 60: Please provide some examples of studies in vegetative tissues, species studied, and cite the references.

Line 65: any differences in vegetative tissues and developing pollens?

Line 107: Please clarify what the room temperature was here in this study. Was that 25oC?

Line 109-112: Any reason to use 30 and 35oC? Any references to support this?

Line 170-171: Please cite references here showing the involvement of osmotin and dehydrins  in stress responses.

Line 247: provide references on heat effects on cytoskeleton.

Line 365 and 366: cite references to support the statement in line 365 and point out studies mentioned in line 366.

Figure 11. nice images. Are these representative images under different treatments? Please also indicate the total number of samples imaged in the legend.

Discussion paragraph 1: this paragraph mainly provided the background information to show the importance of this study. For discussion, the authors can point out the important and novel findings of their study.

Discussion in general: it is hard to tell the results of the present study and from references. Please

Line 411: “not optimal temperature” is a vague term, please clarify or define it.

Line 496 to 503: The conclusion summarized the major results of this study well. However, some expanded discussion on the implications of the results on crop breeding and plant responses to climate warming could be useful. Also, further research perspectives could be suggested based on the present study.

Line 508: please specify the room temperature.

Author Response

Reply to reviewer 3

Mareri et al., Insights Into the Mechanisms of Heat Priming and Thermotolerance in Tobacco Pollen

In this paper, the authors used tobacco pollens to study the effects of heat priming and heat stress on pollen viability, germination, pollen tube length, pollen viability, Calcium level, ROS, organization of actin filaments, cytoskeletal proteins, sugars, and sucrose synthase.  This is a very detailed study with different aspects related to pollen thermotolerance. As the authors mentioned, the effects of priming on pollen grains have been rarely studied. Thus, the data presented here are an important contribution to the thermal tolerance of pollen and have important implications for improving plant thermal tolerance. The manuscript is also generally well-written. In some places, the clarity of the writing could be improved. For example, it is unclear in the discussion that which are the results/findings from the present study, and which are from the literature. So I would suggest clarifying those. The language could also be carefully checked by a native English speaker. The following are some detailed comments:

Line 49: Cite references to support the statement that pollen tubes are very sensitive to high temperatures.

Line 60: Please provide some examples of studies in vegetative tissues, species studied, and cite the references.

Line 65: any differences in vegetative tissues and developing pollens?

Line 107: Please clarify what the room temperature was here in this study. Was that 25oC?

Line 109-112: Any reason to use 30 and 35oC? Any references to support this?

Line 170-171: Please cite references here showing the involvement of osmotin and dehydrins in stress responses.

Line 247: provide references on heat effects on cytoskeleton.

Line 365 and 366: cite references to support the statement in line 365 and point out studies mentioned in line 366.

Figure 11. nice images. Are these representative images under different treatments? Please also indicate the total number of samples imaged in the legend.

Discussion paragraph 1: this paragraph mainly provided the background information to show the importance of this study. For discussion, the authors can point out the important and novel findings of their study.

Discussion in general: it is hard to tell the results of the present study and from references. Please

Line 411: “not optimal temperature” is a vague term, please clarify or define it.

Line 496 to 503: The conclusion summarized the major results of this study well. However, some expanded discussion on the implications of the results on crop breeding and plant responses to climate warming could be useful. Also, further research perspectives could be suggested based on the present study.

Line 508: please specify the room temperature.

We are grateful to the reviewer for the particular attention to our paper. We checked the whole manuscript and added details whenever required; we paid specific attention to the mentioned sentences and mistakes and we addressed the reviewer’s concerns. In particular, main changes performed in the paper are listed below:

Without extending excessively introduction and discussion, recent references have been added in the text in order to make it more comprehensive, updated and intriguing as suggested by the referee. Moreover, we acknowledge that there was ambiguity about the presentation of new data and the citation of bibliography data; therefore, several paragraphs have been rewritten to clarify the difference.

The language has been revised in the whole MS, paying specific attention to the abovementioned suggestions. Moreover, grammatical tenses, prepositions, abbreviations, singular/plural use, etc. have been checked and several parts have been rewritten.

  • During the reediting of discussion, both introductory and concluding paragraphs have been added, in order to highlight the novelty of reported data and to add future perspectives.
  • We removed all ambiguous terms, and we specified the temperatures used, adding the degrees, and clarifying why those temperature was chosen. This info was already present, in the M&M section, which is at the end of the manuscript. For this reason, the choice of 25 and 35 °C (previously already published, study from our lab) is reported also at the beginning of Results in the new version of the manuscript.
  • We specified that we are showing representative images in all the figures containing microscope images (Fig 7, 9, 11)
